# Measuring liquidity in Indian stock market: A dimensional perspective

**Priyanka Naik**[ID], **B. G. Poornima**[ID], **Y. V. Reddy**[ID]*

Goa Business School, Goa University, Taleigao, Goa, India

* yvreddy@unigoa.ac.in

## Abstract

Market liquidity ensures the marketability of security and is an indispensable feature of stock markets. Previous studies have emphasized the role of stock market liquidity in empirical finance. However, they have inadequately explored its multidimensional nature. This study eliminates the ambiguities related to market liquidity by precisely measuring it by using popular and proven liquidity measures. As such, the present study aims to evaluate market liquidity in terms of depth, breadth, tightness, and immediacy in the Indian equity market and also identifies crucial interdependencies between liquidity dimensions. The study selects 500 stocks constituting the NIFTY 500 index of the National Stock Exchange, India, as of 26th May 2019. The data on trading volume, bid price, ask price, the number of shares outstanding, closing share prices were retrieved for the period from 1st April 2009 to 31st March 2019. The study employs Share Turnover, Amihud Illiquidity Ratio, Relative Quoted Spreads, and Coefficient of Elasticity of Trading for liquidity measurement. The Vector Auto-Regressive (VAR) model is used to establish the simultaneous relationships between liquidity dimensions. The analysis is conducted at the aggregate market level as well as across turnover based stock groups divided based on their rankings in terms of stock specific share turnover. The empirical results evidenced the presence of consistent depth, strong breadth, and immediacy but lower tightness in the Indian equity market. The market depth and tightness appear to be relevant in determining dimensional interdependencies. Also, less frequently traded stocks exhibit higher illiquidity in the wake of lower tightness. The findings of this study will assist the investors to wisely understand the multifaceted nature of market liquidity and base their trading decisions accordingly. Moreover, the regulators of the stock exchange can devise liquidity enhancing policies based on the directional movements among liquidity dimensions.

## Introduction

Market liquidity is referred to as the marketability of security [1] and is an essential component of equity markets. It generally refers to the ease with which a security can be exchanged at a given price. A consistent level of market liquidity is of relevance to the market participants, firms and regulators since it ensures continuity in trade at desired prices, regulates the cost of raising capital and allows frictionless functioning of the equity market.

**Data Availability Statement:** Data cannot be shared publicly because of it requires authorized Institutional access. Data are available from the CMIE ProwessIQ and Bloomberg Institutional Data Access (contact via https://prowessiq.cmie.com/

and https://www.bloomberg.com/professional/solution/bloomberg-terminal/) for researchers who meet the criteria for access to data.

**Funding:** The author(s) received no specific funding for this work.

**Competing interests:** The authors have declared that no competing interests exist.

Being an increasingly researched area in finance in the recent last three decades, market liquidity is regarded as an ambiguous concept and has been described in multiple ways by the existing literature. Panayides et al. [2] define market liquidity as trading in security at a lower cost in relation to its actual worth, whereas Hasbrouck and Schwartz [3] refer to it as the immediacy in the execution of a trade. On the other hand, Liu [4] considers market liquidity as the ability to execute a trade in large quantities of security without any delay in time and having no major influence on its price. Additionally, Kyle [5] mentions that liquidity of a market can be understood in terms of three facets namely; the quantity of securities that are traded (depth), the ability of the security prices to quickly recover after a liquidity shock (resiliency) and the costs incurred in trading security (tightness). Furthermore, the time taken to execute a trade (immediacy) and the intensity of trading volume impact on security prices (breadth) is also regarded as the additional facets of market liquidity [6].

The significance of stock market liquidity is highly evident in the asset pricing literature. Studies [1, 7–10] illustrate a pronounced effect of market liquidity on the stock prices and considers it as a crucial component in determining the expected returns for security. Eventually, market liquidity serves as an important attribute in trading strategies, portfolio formulation, and accurate forecasting of portfolio returns [11, 12]. Also, an adequate amount of market liquidity enhances the flow of security-specific information among the market participants and thus promotes efficiency and stability in the stock market [13–15].

Though the empirical works suggest market liquidity as an inevitable component of equity markets but have not been adequately examined with respect to its multidimensional nature. The current context of research has been mainly focused on evaluating liquidity in developed equity markets over a short time period by employing intraday data. Moreover, the recent global financial crisis in the year 2008 has emphasized the role of market liquidity in empirical finance across diverse market structures and time periods, which further makes it necessary to create a fundamental understanding of the different constituents of aggregate market liquidity. Thus, the current study aims at the measurement of liquidity of the Indian stock market by taking into account the different dimensions of liquidity and examines the nature of their interactions with one another over an extended time period. The rationale for selecting the Indian stock market is that it is one of the fastest-growing emerging markets and is more vulnerable to undue market volatilities and market inefficiencies, which in turn necessitates the formulation of suitable policies for market stabilization to facilitate consistent growth. Hence, it is worthwhile to comprehensively perceive the essence of liquidity in the context of the Indian equity market, which will enable in correcting such market imperfections.

The current study measures market liquidity across four dimensions, namely; tightness, depth, breadth, and immediacy over a period of ten years and use daily frequency data. The analysis is performed at the aggregate market level as well as across turnover based quintile groups. By using multiple liquidity measures and Vector Auto-Regressive (VAR) approach, the study exhibits intriguing trading patterns among the liquidity dimensions and prioritizes the role of depth and tightness in the Indian stock market. The study finds that the Indian stock market is characterized by higher persistency in depth, breadth and immediacy but shows lower tightness. The results also reveal that high turnover stocks have a consistent level of liquidity in comparison to low turnover stocks.

Notably, a major challenge while undertaking this study was the selection of appropriate liquidity measures across the four dimensions since the previous literature propounds numerous liquidity measures and their variants. This was resolved by selecting widely acknowledged liquidity measures through extensive analysis of a pool of literary works published on market liquidity. This study contributes in two ways: Firstly, it fruitfully adds to the existing scarce literature on liquidity measurement based on daily data with reference to an order-driven

market. Secondly, the dimensional behavior of liquidity reveals crucial patterns that can be duly considered while devising effective regulatory policies and investment strategies. Moreover, this study offers a novel approach of inclusively studying market liquidity by virtue of widely accepted liquidity measures in an order-driven market system. Further, estimating interdependencies between liquidity dimensions by using daily frequency data and emphasizing the patterns formed thereon is a distinct attempt made with reference to an equity market and will enable traders to effectively optimize their trading strategies.

The entire paper proceeds as follows: the second section provides an overview of related articles on the measurement of stock market liquidity, identifies research gap and highlights objectives of the study; the third section describes data, selected liquidity dimensional measures and methodology; the fourth section provides a discussion of empirical results; the fifth section showcases the conclusions, implications of the study and avenue for future research.

## Review of literature

The nature of market liquidity has been differently conceptualized in the literature, and thus an accurate measurement of market liquidity has been a difficult task for the earlier researches. Black [16] provided distinct characteristics of market liquidity. He mentioned that a liquid market is the one that ensures continuous trading of any quantity of securities at prices close to their current market price in a relatively short time. Based on this, Kyle [5] structured three dimensions of market liquidity i.e., Tightness–cost involved in executing a transaction, depth–the quantity of a security that can be traded without influencing the market price, resiliency–the speed with which prices revert to normal after a shock. In addition to this, the immediacy and breadth aspect of market liquidity was mentioned by Sarr and Lybek [6] which referred to the speed with which a trade can be executed in the market and the extent of price impact caused by trading volume.

Due to its multidimensional nature, market liquidity has been examined by employing numerous measures. The measures tracing the depth and breadth dimension of market liquidity has been popularly employed in comparison to the other dimensions [7, 10, 17–23]. Moreover, liquidity measurement by considering the spread dimension has been adequately examined [1, 3, 24–27], whereas the immediacy dimension is implicitly covered by Wanzala [28] and Wyss [29]. Lastly, the resilience dimension of liquidity has also been investigated [30–33] and is being considered as a complex dimension due to insufficient representation through a single measure [34].

Further, in order to obtain a broader view of market liquidity, many of the recent studies have used a combination of liquidity dimensions in different contexts [35–43]. However, very few studies have tried to uniquely analyze the patterns and relationships between the various dimensions of stock market liquidity. Studies on this matter have mainly concentrated on highlighting the intraday movements in liquidity by employing high-frequency data. Chordia et al. [35] traced a negative correlation between volume, trading activity, and spread liquidity measures, which were mainly observed in the case of large-sized stocks. Hmaied et al. [32] and Olbrys and Mursztyn [44] observed negative relationship between spread and depth measures of stock liquidity as evidenced by vector autoregression approach whereas Ranaldo [34] found a positive association between depth, resiliency and immediacy dimension of market liquidity and observed a U shaped pattern in the liquidity measures at the beginning and end of the trading session.

Additionally, studies have also examined liquidity dimensions by using low-frequency measures. Sklavos et al. [45] analyzed the relationship between spread, turnover, and price impact liquidity measures in the context of the energy sector. By following structural vector autoregression, they found a consistent interrelationship between all measures and obtained significant causality led by turnover to the spread and price impact measures. Daníelsson and Payne

[46] identified a positive association between depth, spread, and trading activity measures but found that this relationship disappears during high volatility. Yeyati et al. [47] found a negative relation between depth and spread measures and evidenced that the relationship inverses during the period of crisis.

In the context of the Indian stock market, Krishnan and Mishra [48] used spread, depth, and multifaceted liquidity measures and found that there is high liquidity at the beginning and end of the trading sessions. The study also established a positive relationship between depth and spread measures and by using Principal Component Analysis evidenced moderate intra-day movement between the measures. Jha et al. [49] examined the liquidity using BSE 500 stocks and concluded that Indian market liquidity gets adversely affected by financial shocks, but at the same time, it was observed that it normalizes within a short time duration thereby depicting high market resilience. Bhattacharya et al. [50] employed the Autoregressive Distributed-lag (ARDL) Bounds Testing Approach and perceived that market liquidity measured in terms of depth, immediacy, tightness and resiliency exhibit a significant long term relationship with the overall Indian stock market as proxied by BSE 500 index.

The existing literature provides multiple dimensions for determining the market liquidity wherein Chai et al. [19] and Goyenko et al. [51] empirically suggests that each of these dimensions captures distinct aspects of market liquidity in different market systems and conditions. But most studies have not adequately analyzed all the liquidity dimensions in a single study. Also, the literature has rarely emphasized on capturing the interdependencies between the liquidity dimensions that can literally help in framing coherent trading strategies. Moreover, less attention has been paid towards determining liquidity based on low-frequency data. Kim and Lee [39] suggest that measures based on low-frequency data enable to study liquidity over a long time span and across different market structures. However, a major gap in the literature is the lack of empirical work on studying market liquidity in the context of order-driven markets. These markets substantially offer diversified investment opportunities to a wide range of investors and are strongly prone to liquidity shocks since consistency in trading activity plays a pivotal role in supplying liquidity due to the absence of designated liquidity providers. Thus the foremost understanding of the level and characteristics of liquidity in such markets becomes imperative.

The current study aims to fill these gaps by adequately measuring the liquidity of the Indian stock market by using different dimensional measures and examine the interdependencies between the dimensional measures of liquidity for the overall market as well as with regards to variably traded stocks. The present study will assist the investors to thoroughly understand the multifaceted nature of market liquidity in the Indian stock market and base their trading decisions accordingly. Even the regulators and stock exchanges can devise constructive policies with an intention to preserve and enhance liquidity in the market based on the directional movements among liquidity dimensions.

## Materials and methods

### Data

For the purpose of the study, we select 500 stocks constituting the NIFTY 500 index of the National Stock Exchange, India, as of 26th May 2019. The data on trading volume, bid price, ask price, the number of shares outstanding, closing share prices were retrieved for the period from 1st April 2009 to 31st March 2019 from Bloomberg and Centre for Monitoring Indian Economy (CMIE) Prowess database on a daily basis. It was necessary that this collected data was available throughout the study period across the selected stocks, and thus only 352 stocks qualified to be included in the final sample.

We measure market liquidity in terms of four dimensions, as presented by Sarr and Lybek [6] namely, Tightness, Immediacy, Depth, and Breadth. Moreover, the study does not consider the fifth dimension of resiliency due to inadequate empirical support regarding acceptable resiliency measures with reference to order-driven markets. The previous literature has provided a wide range of measures for tracing every dimension of liquidity. By analyzing the same, we select the widely applied low-frequency measures which enable in the adequate capture of every liquidity dimension over an extended period of time. However, the selection of measures was restricted to the accessibility of the required data for its computation in the available databases.

Next, the study first defines four liquidity dimensions and also describes the selected measures of liquidity to be used for the purpose of analysis:

**Tightness.**   Tightness indicates the amount of cost incurred by an investor for transacting security. The Bid-Ask Spreads has been mainly used in the earlier studies to evaluate the tightness in the market. But Grossman and Miller [52] state that in a market, the trades may not always be executed at quoted Bid-Ask prices, and thus it becomes inappropriate to determine the real transaction cost. In consideration of this, we use Relative Quoted Spread as a measure of tightness as suggested by Yilmaz et al. [53], and Foran et al. [54], which expresses daily spread as a percentage of the midpoint of the bid and ask prices. It is calculated as follows:

$$Relative\ Quoted\ Spread = \frac{AP_{it} - BP_{it}}{1/2}(AP_{it} + BP_{it}) \tag{1}$$

where AP and BP denote daily closing Ask Price and daily closing Bid Price; i and t denote stock i at time t. A narrow Relative Quoted Spread would indicate lower transaction costs and higher liquidity on account of a tighter market.

**Immediacy.**   Immediacy indicates the execution time required for a transaction, which depends on the willingness of both the parties to a transaction to execute the stated quantity of a security at the quoted price without any delay in time. In order to measure market immediacy, the study uses the Coefficient of Elasticity of Trading as suggested by Wanzala [28] which rightly depicts the speed of execution of a trade as depicted through percentage change in trading quantity for a percentage change in the share price. This measure is calculated as follows:

$$Coefficient\ of\ Elasticity\ of\ Trading = \frac{\%\Delta T_s}{\%\Delta P} \tag{2}$$

Where $\%\Delta T_s$ denotes the percentage change in the daily trading volume of a stock 's' and $\%\Delta P$ denotes the percentage change in daily closing price. A larger Coefficient of Elasticity of Trading indicates a higher immediacy and thereby confirms higher liquidity.

**Depth.**   Depth refers to the availability of a fairly large amount of orders in the market such that it maintains equilibrium in security's market price. Thus, the amount of security traded in the market is a prerequisite for a deep market to exist and can be easily ascertained through the trading volume. Since the study employs stocks of varied sizes to determine market liquidity, it uses the liquidity measure of Share Turnover as it considers volume traded as a proportion to the number of shares outstanding [55]. It is computed as follows:

$$Share\ Turnover = \frac{VO_t}{SO_t} \tag{3}$$

Where $VO_t$ is the number of shares traded on day t and $SO_t$ is the number of shares outstanding on day t. A higher amount of share turnover indicates a deep market and represents higher liquidity.

**Breadth.** Breadth refers to the ability of the market to smoothly enable trading of a given volume of securities without much influencing the share prices. Price impact measures are effectively used in the context of evaluating the liquidity dimension of breadth. We use the Amihud Illiquidity Ratio as proposed by Amihud [7] for measuring the market breadth and is regarded as the best price impact measure by Goyenko et al. [51]. This ratio exhibits the movement in the security prices due to changes in its volume and is computed as follows:

$$\text{Amihud Illiquidity Ratio} = \frac{|R_{it}|}{Vol_{it}} \qquad (4)$$

Where $|R_{it}|$ and $Vol_{it}$ are the absolute return and volume (in Rs.) on day t for stock i respectively. A lower Amihud Illiquidity Ratio portrays the wide market breadth and thereby conveys the presence of high liquidity.

## Methodology

Each of the selected liquidity measures was calculated on a daily basis for every stock constituting the final sample over the entire study period. Non-positive Relative Quoted Spreads were eliminated as suggested by Sklavos et al. [45] and Jacoby and Zheng [56]. In order to derive the aggregate market liquidity, the daily stock-specific measures were aggregated into cross-sectional averages that were weighted based on the daily market capitalization of these stocks and were further averaged across time to derive monthly averages. Next, to avoid any form of outliers, these monthly averages of liquidity measures were converted into natural log values [57–59]. Lastly, these log values were evaluated to derive the results relating to aggregate market liquidity.

Further, to assess interdependency among the liquidity dimensions, the study employs the Vector Auto-Regressive (VAR) model. The study models Relative Quoted Spread (RQS), Coefficient of Elasticity of Trading (CET), Amihud Illiquidity Ratio (AR), and Share Turnover (ST) using VAR as follows:

$$
\begin{bmatrix} CET_t \\ AR_t \\ RQS_t \\ ST_t \end{bmatrix} + 
\begin{bmatrix} \alpha_1\alpha_2\alpha_3\alpha_4\alpha_5\alpha_6\alpha_7\alpha_8\alpha_9\alpha_{10}\alpha_{11}\alpha_{12} \\ \beta_1\beta_2\beta_3\beta_4\beta_5\beta_6\beta_7\beta_8\beta_9\beta_{10}\beta_{11}\beta_{12} \\ \theta_1\theta_2\theta_3\theta_4\theta_5\theta_6\theta_7\theta_8\theta_9\theta_{10}\theta_{11}\theta_{12} \\ \delta_1\delta_2\delta_3\delta_4\delta_5\delta_6\delta_7\delta_8\delta_9\delta_{10}\delta_{11}\delta_{12} \end{bmatrix}
\begin{bmatrix} c \quad c \quad c \quad c \\ CET_{t-1}CET_{t-1}CET_{t-1}CET_{t-1} \\ CET_{t-2}CET_{t-2}CET_{t-2}CET_{t-2} \\ AR_{t-1}AR_{t-1}AR_{t-1}AR_{t-1} \\ AR_{t-2}AR_{t-2}AR_{t-2}AR_{t-2} \\ RQS_{t-1}RQS_{t-1}RQS_{t-1}RQS_{t-1} \\ RQS_{t-2}RQS_{t-2}RQS_{t-2}RQS_{t-2} \\ ST_{t-1}ST_{t-1}ST_{t-1}ST_{t-1} \\ ST_{t-2}ST_{t-2}ST_{t-2}ST_{t-2} \\ CET_tCET_tCET_tCET_t \\ AR_tAR_tAR_tAR_t \\ RQS_tRQS_tRQS_tRQS_t \\ ST_tST_tST_tST_t \end{bmatrix}
+ \begin{bmatrix} \varepsilon_{1t} \\ \varepsilon_{2t} \\ \varepsilon_{3t} \\ \varepsilon_{4t} \end{bmatrix} \qquad (5)
$$

The coefficients $\alpha_{10}$, $\beta_{10}$, $\theta_{11}$ and $\delta_{12}$ are set to zero to assess the causality impact on the other variables in the system. In addition, the liquidity dimensions and their interdependencies are also evaluated across low and high traded stocks by equally dividing the entire sample

stocks into five Quintiles based on their rankings in terms of stock specific share turnover at the beginning of each year wherein 1st Quintile (also referred as upper quintile) comprised of highly traded stocks while 5th Quintile (also referred as lower quintile) consisted of less traded stocks. Each quintile had a total of 70 stocks (except 5th Quintile that consisted of 72 stocks) throughout the sample period, and the constituting stocks in every quintile were allowed to vary based on changes in their turnover based on rankings every year.

From the VAR Model, we expect crucial interdependencies between the selected liquidity dimensions at the market level as well as across the Quintile groups. Diaz and Escribano [60] suggests that depth and breadth dimensions tend to be interrelated as they both rely on the level of volume exchanged for security. Sklavos et al. [45] evidenced that market tightness resembles risk to a market maker that gets influenced by market depth and breadth but conversely, they are not affected by market tightness. On the other hand, liquidity in an order-driven market is supplied by the traders themselves and thus market tightness, depth, breadth, and the resultant immediacy [60] tend to be interdependent at various time lags due to the presence of different types of traders. As evidenced by the existing literature, a liquid market is characterized to consist high level of tightness, immediacy, depth, and breadth and hence with this consideration, the study assumes that there exists a positive interdependency between market tightness, immediacy, depth, and breadth.

## Results and discussion

### Descriptive statistics

The descriptive statistics of the liquidity measures are given in Table 1. They indicate nominal values over the study period. The mean values depict the higher value of CET, and lower value of AR in comparison to other measures, thereby indicating that large volumes of security can be easily traded in the market without any delay in time at a lower price impact. Further, the RQS is higher in comparison to ST and AR thus exhibits higher trading costs for executing a transaction in the market. Even then, ST has been consistent (as indicated by lower standard deviation) over the period and has stabilized the volume induced price fluctuations and enhanced immediacy. Among the liquidity measures, CET exhibits the highest standard deviation, whereas ST has the lowest variation over the study period. The liquidity dimensions exhibit positive skewness, and their mean values are very near to the median.

Table 2 gives descriptive statistics of liquidity measures across the Quintiles created based on share turnover rankings at the beginning of each year. It exhibits a similar pattern as obtained in the case of the full sample, wherein the value of AR has been the lowest, and that of CET is the highest; also, ST has been consistent and is lower than RQS. Additionally, most of the values are positively skewed.

**Table 1. Descriptive statistics of the liquidity measures for the full sample.**

|  | AR | CET | RQS | ST |
|---|---|---|---|---|
| **Mean** | -11.09351 | -0.101038 | -4.628430 | -4.741760 |
| **Median** | -11.09536 | -0.100795 | -4.621878 | -4.748665 |
| **Maximum** | -10.79229 | 0.190973 | -4.470929 | -4.600180 |
| **Minimum** | -11.28886 | -0.371318 | -4.698233 | -4.785492 |
| **Std. Dev.** | 0.042093 | 0.073298 | 0.038100 | 0.030615 |
| **Skewness** | 2.416088 | 0.165493 | 0.673896 | 2.256283 |
| **Kurtosis** | 27.27266 | 6.758769 | 4.343550 | 9.778872 |

Data from databases of CMIE, Bloomberg, and authors' calculations.

**Table 2. Descriptive statistics of the liquidity measures for quintile groups.**

| | 1st Quintile | | | | 2nd Quintile | | | | 3rd Quintile | | | |
|---|---|---|---|---|---|---|---|---|---|---|---|---|
| | AR | CET | RQS | ST | AR | CET | RQS | ST | AR | CET | RQS | ST |
| Mean | -10.156 | 0.339 | -3.992 | -3.606 | -11.965 | 0.403 | -4.044 | -4.056 | -10.490 | 0.454 | -4.011 | -4.211 |
| Median | -10.160 | 0.335 | -3.987 | -3.615 | -11.943 | 0.408 | -4.040 | -4.059 | -10.489 | 0.480 | -4.007 | -4.217 |
| Maximum | -9.858 | 0.631 | -3.855 | -3.467 | -11.607 | 0.699 | -3.913 | -3.970 | -10.186 | 0.767 | -3.873 | -4.094 |
| Minimum | -10.227 | -0.150 | -4.066 | -3.671 | -12.918 | -0.075 | -4.124 | -4.124 | -10.615 | -0.647 | -4.082 | -4.273 |
| Std. Dev. | 0.037 | 0.093 | 0.048 | 0.037 | 0.132 | 0.108 | 0.042 | 0.033 | 0.033 | 0.169 | 0.042 | 0.042 |
| Skewness | 7.185 | -0.921 | 0.312 | 1.129 | -3.440 | -0.854 | 0.460 | 0.226 | 5.911 | -3.864 | 0.392 | 0.629 |
| Kurtosis | 55.536 | 10.580 | 2.522 | 4.597 | 25.237 | 7.456 | 3.248 | 2.691 | 64.084 | 24.158 | 3.030 | 2.812 |
| | 4th Quintile | | | | 5th Quintile | | | | | | | |
| | AR | CET | RQS | ST | AR | CET | RQS | ST | | | | |
| Mean | -10.769 | 0.608 | -3.749 | -4.433 | -9.351 | 0.501 | -3.712 | -4.637 | | | | |
| Median | -10.737 | 0.612 | -3.751 | -4.440 | -9.355 | 0.564 | -3.721 | -4.634 | | | | |
| Maximum | -10.429 | 0.890 | -3.546 | -4.320 | -9.053 | 0.848 | -3.561 | -4.501 | | | | |
| Minimum | -11.564 | 0.270 | -3.798 | -4.532 | -9.509 | -0.951 | -3.786 | -4.734 | | | | |
| Std. Dev. | 0.119 | 0.093 | 0.032 | 0.045 | 0.039 | 0.281 | 0.044 | 0.051 | | | | |
| Skewness | -4.356 | -0.441 | 3.240 | 0.374 | 3.410 | -2.334 | 0.736 | 0.241 | | | | |
| Kurtosis | 27.594 | 5.881 | 20.746 | 2.865 | 31.976 | 10.098 | 3.488 | 2.522 | | | | |

Data from databases of CMIE, Bloomberg, and authors' calculations.

Between the upper and lower quintile groups, we observe a huge difference in ST (nearly 29%). Also, AR is lower for the upper quintile, thereby indicates that highly traded stocks generate lower price impact due to continuity in trading of such stocks. Regarding RQS, stocks in the upper quintile witness reduced spreads than those in lower quintiles. This suggests that it is cheaper to transact in highly traded stocks rather than low traded ones. Moreover, CET is higher for low traded stocks, but at the same time, it is highly unstable. Interestingly, we notice that AR and RQS are the lowest in the case of stocks in 2nd Quintile, which means that moderately higher traded stocks have lower trading costs and price impact.

## Unit root test

In order to examine the stationarity of the liquidity measures, the study employs the Augmented Dickey Fuller Test for full sample as well as across the Quintile groups. The ADF unit root results are shown in Table 3. It evidences significant results and hence the null hypothesis is rejected and thus states that all the liquidity measures are stationary.

**Table 3. ADF unit root results of the liquidity measures for the full sample and quintile groups.**

| Null Hypothesis | Full Sample | 1st Quintile | 2nd Quintile | 3rd Quintile | 4th Quintile | 5th Quintile |
|---|---|---|---|---|---|---|
| No unit root in AR | -11.300*** | -11.412*** | -8.133*** | -13.937*** | -11.251*** | -10.647*** |
| No unit root in CET | -9.769*** | -10.271*** | -11.648*** | -10.874*** | -5.809*** | -9.952*** |
| No unit root in RQS | -5.015*** | -4.222*** | -4.394*** | -4.688*** | -8.278*** | -4.424*** |
| No unit root in ST | -5.113*** | -4.495*** | -4.862*** | -3.203** | -2.730* | -4.510*** |

Figures represent the t statistics

***significant at 1%

**significant at 5%

*significant at 10%.

Data from databases of CMIE, Bloomberg, and authors' calculations.

## Correlation analysis

Table 4 shows the correlation between the various liquidity measures employed in the VAR model. It shows a positive correlation between ST and RQS in the full sample as well as across the Quintile groups. However, the strength of this relationship is lower for less traded stocks. This is due to the fact that volume-based trading is quite common in highly traded stocks. Such a form of trading is undertaken with an aim to reap adequate returns from trade which results in an increased impact on security prices and raises volatility in trade and induces the spreads to widen in order to accommodate increased uncertainty [48]. This is also evident from the stocks in the 2nd quintile, which shows that AR is positively correlated with ST and RQS and negatively related to CET. Further, ST is found to be positively related to AR in the case of stocks in the 4th Quintile, which means that the low turnover stocks tend to have lower price impact.

## Vector autoregression estimates

**Estimates for the full sample.** The interactions among the various liquidity measures are tested through VAR and are depicted in Table 5. The total variations in AR, CET, RQS, and ST are explained by the other variables to the extent of 10%, 3%, 68%, and 70%, respectively. AR and ST are significantly affected by both the lags of RQS but in opposite directions. It is evident that CET is exogenously determined as it is not significantly dependent on any of the liquidity measures. On the contrary, CET shows a positive effect on ST and RQS, which indicates that previous day immediacy raises the present values of spreads and share turnover. The reason being that increased speed of execution ensures the presence of willing counterparties to trade in securities and results in a higher trading activity which makes future trades costlier on account of cautious trades. Regarding ST, the estimates indicate that it is mainly characterized by its own previous day lag in comparison to other determining variables. This exhibits that the current trading activity is caused by the past trading activity which seems to be increasing the future spreads as evidenced by the positive effect of one day lag of ST on RQS. This suggests for the presence of informed trading in the market that drives the next day, turnover widens the future spreads and causes an immediate impact on prices [45, 61]. Moreover, RQS displays an effect of its own lag value that hints at the market's adjustment to the uncertainty brought in by informed traders. At the same time, current values of RQS are negatively dependent on two days lag of ST. This can be due to the fact that informed trading persuades more uninformed trading, which contributes to reducing the information asymmetry [62] and

**Table 4. Correlation analysis between liquidity measures for the full sample and quintile groups.**

| Null Hypothesis | Full Sample | 1st Quintile | 2nd Quintile | 3rd Quintile | 4th Quintile | 5th Quintile |
|---|---|---|---|---|---|---|
| No correlation between AR & CET | -0.032 | 0.028 | -0.197** | -0.04 | -0.065 | -0.048 |
| No correlation between AR & RQS | -0.074 | -0.024 | 0.153* | 0.02 | -0.106 | -0.003 |
| No correlation between AR & ST | 0.088 | 0.058 | 0.201** | -0.049 | 0.243*** | -0.039 |
| No correlation between CET & RQS | -0.108 | 0.007 | 0.0106 | -0.033 | -0.054 | -0.039 |
| No correlation between CET & ST | -0.109 | 0.004 | 0.012 | 0.063 | -0.258 | 0.098 |
| No correlation between ST & RQS | 0.429*** | 0609*** | 0.659*** | 0.521** | 0.183** | 0.211** |

Figures represent the coefficients

***significant at 1%

**significant at 5%

*significant at 10%.

Data from databases of CMIE, Bloomberg, and authors' calculations.

**Table 5. VAR results for liquidity measures in the full sample.**

| | C | AR(-1) | AR(-2) | CET(-1) | CET(-2) | RQS(-1) | RQS(-2) | ST(-1) | ST(-2) | $R^2$ |
|---|---|---|---|---|---|---|---|---|---|---|
| **AR** | -10.465*** (1.652) | -0.077 (0.092) | 0.122 (0.092) | -0.065 (0.054) | 0.061 (0.054) | -0.608*** (0.218) | 0.595*** (0.209) | 0.322 (0.294) | -0.281 (0.284) | 0.102 |
| **CET** | 0.335 (2.981) | 0.138 (0.167) | 0.013 (0.166) | 0.103 (0.098) | 0.001 (0.098) | 0.299 (0.393) | -0.333 (0.377) | -0.647 (0.530) | 0.417 (0.512) | 0.037 |
| **RQS** | -1.823** (0.793) | -0.011 (0.044) | 0.011 (0.044) | 0.053** (0.026) | -0.029 (0.026) | 0.711*** (0.105) | 0.107 (0.100) | 0.151* (0.141) | -0.357*** (0.136) | 0.681 |
| **ST** | -2.065*** (0.567) | -0.030 (0.031) | -0.052* (0.032) | 0.036** (0.018) | -0.003 (0.019) | -0.124* (0.075) | 0.129* (0.072) | 0.704*** (0.101) | 0.047 (0.097) | 0.702 |

Figures represent the coefficients; Probability values in brackets

*** significant at 1%

** significant at 5%

* significant at 10%.

Data from databases of CMIE, Bloomberg, and authors' calculations.

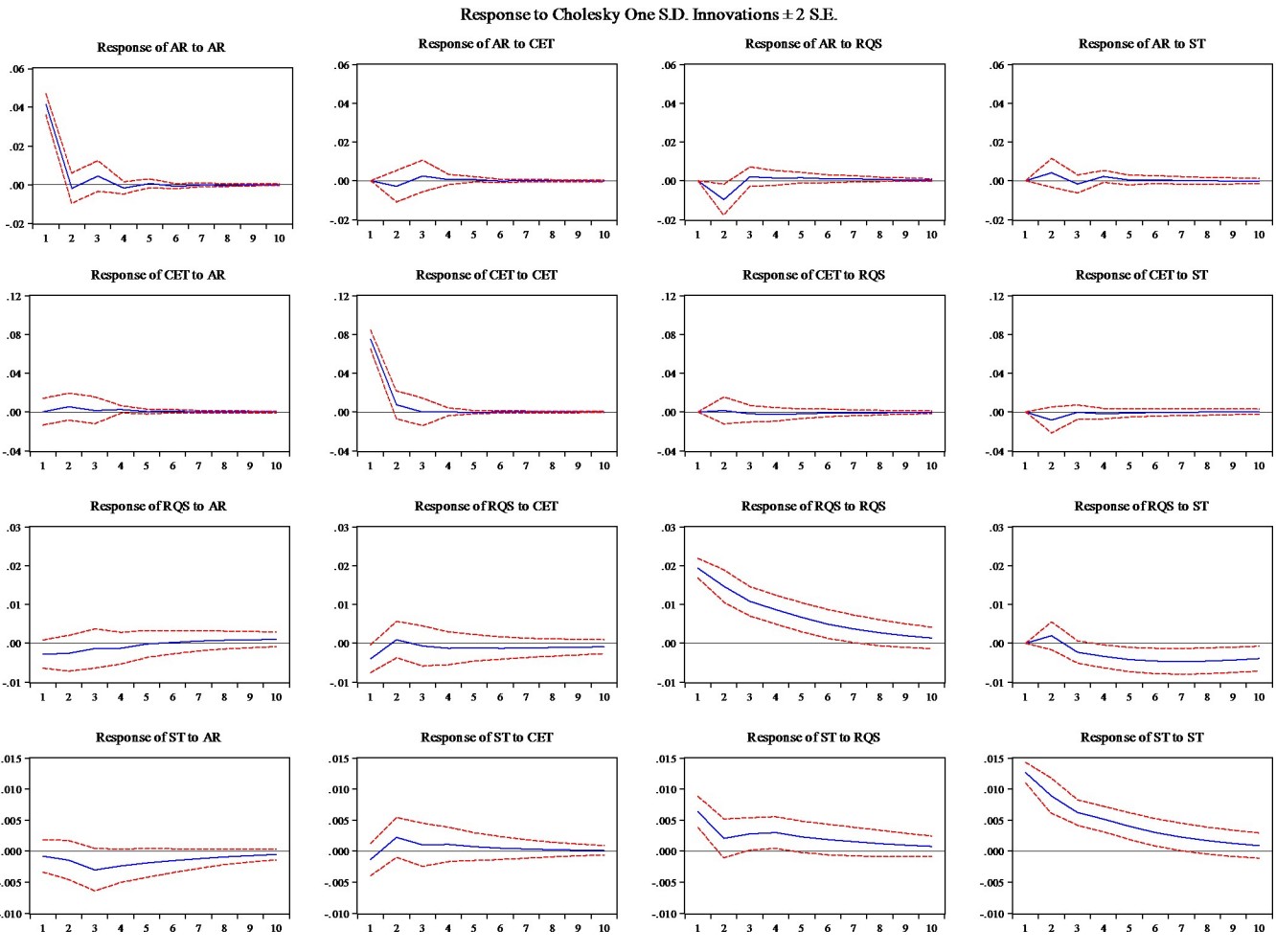

**Fig 1. Impulse Response Function for liquidity measures in the full sample.**

thereby contracts the RQS and stabilizes the price impact. Thus AR and ST witness the opposite effects of RQS at various time lags.

Next, the results of the Impulse Response Function of the full sample are given in Fig 1 and support the patterns observed in the VAR model. The Impulse Responses show the extent of response of a variable to a given change (shock or information) in another variable. In this study, they are plotted for a period of 10 days. It is to be noted that all liquidity measures respond quickly to their own lagged changes. In this context, the response of AR to AR and CET to CET diminishes quickly and reaches equilibrium within 3 observation days, while in the case of RQS to RQS and ST to ST it takes 10 days and shows a concentrated response. Further, it is evident that CET is not dependent on any of the liquidity dimensions and that AR responds negatively to RQS and quickly adjusts within 3 observation days. Also, a long period opposite impact is observed in RQS and ST on account of shocks in each other.

**Estimates for quintile groups.** Table 6 shows the VAR estimates for liquidity measures and their interrelationships for the turnover based quintiles. In the case of 1st Quintile; the two days lag of CET negatively affects AR, whereas RQS and ST are positively dependent. Also, we find that the previous day AR positively affects RQS. This refers that in highly traded stocks; past increased immediacy in trade generates more trading activity, lowers the impact on prices,

Table 6. VAR results for liquidity measures in quintile groups.

| | C | AR(-1) | AR(-2) | CET(-1) | CET(-2) | RQS(-1) | RQS(-2) | ST(-1) | ST(-2) | $R^2$ |
|---|---|---|---|---|---|---|---|---|---|---|
| **1st Quintile** | | | | | | | | | | |
| AR | -10.565*** (1.330) | -0.0429 (0.089) | -0.044 (0.090) | 0.022 (0.036) | -0.145*** (0.035) | 0.088 (0.134) | -0.114 (0.134) | 0.267 (0.186) | -0.118 (0.182) | 0.160 |
| CET | -2.177 (3.558) | -0.257 (0.238) | -0.122 (0.240) | 0.052 (0.095) | -0.126 (0.094) | 0.071 (0.359) | 0.080 (0.359) | 0.309 (0.497) | -0.115 (0.486) | 0.046 |
| RQS | 0.311 (0.959) | 0.123** (0.064) | 0.018 (0.065) | 0.029 (0.026) | 0.066*** (0.025) | 0.592*** (0.097) | 0.247*** (0.097) | 0.355*** (0.134) | -0.476*** (0.131) | 0.698 |
| ST | -1.223* (0.706) | 0.013 (0.047) | -0.047 (0.048) | 0.007 (0.019) | 0.060*** (0.019) | -0.064 (0.071) | 0.124* (0.071) | 0.839*** (0.098) | -0.142 (0.096) | 0.688123 |
| **2nd Quintile** | | | | | | | | | | |
| AR | -5.628*** (1.828) | 0.229*** (0.096) | -0.097 (0.093) | 0.061 (0.111) | 0.008 (0.112) | -0.332 (0.500) | 0.499 (0.486) | 0.451 (0.543) | 0.562 (0.553) | 0.156 |
| CET | 0.008 (1.622) | 0.001 (0.085) | 0.005 (0.083) | -0.093 (0.099) | -0.076 (0.099) | -0.122 (0.444) | -0.120 (0.431) | 0.669 (0.482) | -0.561 (0.490) | 0.039 |
| RQS | -0.815** (0.373) | -0.016 (0.020) | 0.043** (0.019) | 0.006 (0.023) | 0.021 (0.023) | 0.656*** (0.102) | 0.065 (0.099) | 0.203* (0.111) | -0.203* (0.113) | 0.616 |
| ST | -1.360*** (0.344) | -0.000 (0.018) | 0.001 (0.017) | 0.041** (0.021) | 0.041** (0.021) | -0.039 (0.094) | 0.232*** (0.091) | 0.594*** (0.102) | -0.116 (0.104) | 0.509 |
| **3rd Quintile** | | | | | | | | | | |
| AR | -15.013*** (1.598) | -0.281*** (0.093) | -0.113 (0.092) | 0.007 (0.017) | 0.027 (0.018) | -0.151 (0.137) | 0.219** (0.127) | 0.078 (0.145) | -0.233* (0.141) | 0.134 |
| CET | 14.614* (8.491) | 0.392 (0.493) | 0.974*** (0.491) | -0.026 (0.094) | -0.063 (0.098) | -0.107 (0.730) | -0.684 (0.677) | -0.319 (0.772) | 1.022 (0.747) | 0.069 |
| RQS | -0.751 (1.312) | -0.013 (0.076) | 0.032 (0.076) | -0.004 (0.014) | -0.004 (0.015) | 0.646*** (0.113) | 0.089 (0.105) | 0.127 (0.119) | -0.010 (0.115) | 0.588 |
| ST | -0.336 (1.224) | 0.008 (0.071) | 0.035 (0.071) | 0.034*** (0.013) | 0.001 (0.014) | -0.050 (0.105) | 0.016 (0.097) | 0.752*** (0.111) | 0.098 (0.108) | 0.684 |
| **4th Quintile** | | | | | | | | | | |
| AR | -9.234*** (2.011) | 0.095 (0.090) | 0.091 (0.087) | 0.048 (0.111) | 0.117 (0.110) | 0.564 (0.623) | -1.072** (0.491) | 0.529 (0.362) | -0.183 (0.355) | 0.112 |
| CET | -0.262 (1.701) | -0.126* (0.076) | 0.094 (0.074) | -0.002 (0.094) | 0.190** (0.093) | 0.228 (0.527) | 0.240 (0.415) | -0.194 (0.306) | -0.295 (0.301) | 0.135 |
| RQS | -1.403*** (0.299) | -0.004 (0.013) | 0.031*** (0.013) | -0.006 (0.016) | -0.013 (0.016) | 0.431*** (0.093) | 0.094 (0.073) | 0.056 (0.054) | -0.039 (0.053) | 0.463 |
| ST | -0.497 (0.543) | 0.031 (0.024) | -0.008 (0.024) | 0.033 (0.030) | 0.024 (0.030) | -0.048 (0.168) | 0.032 (0.133) | 0.571*** (0.098) | 0.282*** (0.096) | 0.629 |
| **5th Quintile** | | | | | | | | | | |
| AR | -9.178*** (1.280) | -0.024 (0.090) | 0.060 (0.091) | -0.002 (0.013) | -0.001 (0.013) | -0.247* (0.151) | 0.310* (0.144) | 0.077 (0.103) | -0.163* (0.100) | 0.079 |
| CET | 7.068 (9.589) | 0.684 (0.678) | 0.060 (0.680) | 0.071 (0.096) | 0.090 (0.095) | -0.204 (1.135) | -0.244 (1.075) | 0.796 (0.771) | -0.504 (0.753) | 0.036 |
| RQS | -1.546** (0.803) | -0.055 (0.057) | -0.052 (0.057) | 0.009 (0.008) | -0.007 (0.008) | 0.762*** (0.095) | -0.034 (0.090) | 0.070 (0.064) | 0.030 (0.063) | 0.666 |
| ST | -0.169 (1.190) | 0.088 (0.084) | -0.033 (0.084) | 0.001 (0.012) | -0.007 (0.012) | 0.183 (0.141) | -0.026 (0.133) | 0.558*** (0.096) | 0.167* (0.093) | 0.533 |

Figures represent the coefficients; Probability values in brackets

***significant at 1%

**significant at 5%

*significant at 10%.

Data from databases of CMIE, Bloomberg, and authors' calculations.

and leads to an increase in the trading costs and vice versa. It is observed that ST is dependent on its time lag, which means that investors base their trading behavior by referring to the past trading activity, thereby confirms the notion of the information content of turnover. Also, RQS is positively affected by its lagged values and also by one day lag of ST. This depicts that trading costs and trading activity of the earlier periods define the crucial movements in present-day costs. This mainly arises when informed traders desire to immediately materialize the benefit of their informational advantage [63] and thus a large increase in their trading activity raises the cost of immediate execution. Moreover, a higher intensity in trading smoothens the flow of information among all market participants and thereby enables optimal trading, which further eases the trading costs. This is evident from the negative effect of two days ST on RQS. In 2nd Quintile; AR depends on its previous day lag, whereas interrelationships between ST, CET, and RQS are similar as found in 1st Quintile.

On the other hand, the 3rd, 4th, and 5th Quintiles describe a similar level of interdependencies between the liquidity measures. The results show that huge variations in RQS are solely caused by their own past movements. This indicates the adjustment of trades towards the risk induced by information asymmetry. Moreover, low turnover stocks have infrequent trading activity and lower dissemination of stock-related information. Thus transactions in such stocks will be always executed at a higher cost and will constantly elevate future costs. Additionally, Easley et al. [64] mentions that spreads of low traded stocks are mainly characterized by higher information asymmetry, which causes a wide effect on their prices. Thus an investor with relevant information can strategize their trades and fulfill their profit motives but relevantly at higher costs. Further, RQS is found to positively affect the present values of AR. Since the increased execution costs will discourage further trades which will in turn result in a higher impact on their prices. It is seen that ST depends on its lagged values, and AR is negatively affected by two days lag of ST. This means that lower turnover of less traded stocks reduces the present trading activity and broaden the price impact. Remarkably, the immediate execution of trades in low quintiles is facilitated by price impact as CET is found to be dependent on one day lagged value of AR which prompts the strong presence of informed trading activity.

Furthermore, S1 Fig depicts the Impulse Response Functions of liquidity measures across the Quintile groups over an observation period of 10 days, which validate the relationships obtained in the VAR model. The Impulse Responses exhibit an instant response to their own shocks wherein the response of AR to AR and CET to CET reaches equilibrium within 4 observation days while it requires 10 days for RQS and ST to absorb their own changes. Though the intensity of the response of liquidity measures to the shocks eases in the lower quintiles but notably the responses of RQS to RQS amplifies and takes a long time to reach parity. This validates the notion that infrequently traded stocks have higher execution risk and are transacted at inflated trading costs.

## Conclusions

The previous literature propounds significance of market liquidity in different contexts of empirical finance but is insufficiently analyzed with regards to multi-dimensional measurement of liquidity and tracing their interdependencies. The current study aims to measure the liquidity of the Indian equity market and investigates the extent of interdependency between different dimensional aspects of market liquidity. The market liquidity is evaluated in terms of four dimensions namely; tightness, immediacy, depth, and breadth, whereas interactions between these dimensions are analyzed by using the Vector Autoregression (VAR) model. The study employs daily frequency data from 1st April 2009 to 31st March 2019 and conducts analysis at the aggregate market level as well as across annual turnover ranked quintile groups.

The study concludes that liquidity in the context of the Indian equity market is characterized by consistent depth, higher breadth, and immediacy but displays a lower tightness at aggregate market as well as group level. Besides, there exists a significant negative relation between depth and tightness (represented by positive relation between ST and RQS) and is observed to be stronger in case of high traded stocks, thereby confirms the existence of volume-based trading in the stock market. These results are in confirmation to those obtained from intraday data series by Krishnan and Mishra [48] in the Indian stock market. Additionally, the study arrives at a unique set of interdependencies between the dimensional measures of liquidity, which are in contrast to the assumptions of the study. It is found that market depth and tightness are mainly dependent on their lagged values which confirms the assumption of informational role of trading activity as suggested by Anderson [65] and Bohl and Henke [66], and asymmetric information component of quoted spreads as concluded by Hasbrouck [24] and Glosten and Harris [67]. Further, the study evidenced that depth and tightness in the market are divergently dependent on different time-lagged values of each other and indicate a pertinent impact on liquidity provision through interactions between informed and uninformed trading patterns. Moreover, immediacy is found to be independently determined in the market except with respect to immediacy in lower quintile stocks, which was perceived to be stimulated by market breadth.

In addition, the study concludes the existence of higher persistence in liquidity for high turnover stocks in comparison to lower turnover stocks as exhibited by higher depth, breadth, and tightness and matches with the findings obtained by Sklavos et al. [45]. Also, the past trading activity in highly traded stocks immensely advances their present depth and stabilizes the undue movements in trading costs, which further contributes in higher liquidity of such stocks. On the contrary, lower depth and tightness in case of less frequently traded stocks significantly impairs their market breadth.

The findings of this study will have important implications for the investors and market regulators in coherent understanding of the essence of market liquidity in the Indian equity market, strategizing of trades and devising constructive policies to boost market confidence. The results of this study suggest that the investors should consider the movements among the market depth and tightness as they are crucial in understanding variations in market liquidity. Also, the investment portfolio should encompass stocks with higher turnover so as to provide a cushion against liquidity shocks. Additionally, due consideration needs to be given by the market regulator at enhancing tightness in low traded stocks so as to lessen extreme share price variations. In future research, the current study can be readily extended by evaluating the impact of information-based trading on market liquidity levels. Even the impact of macroeconomic announcements made by domestic and global markets can also be undertaken for perceiving the systematic co-movements between liquidity dimensions.

## Supporting information

**S1 Fig. Impulse Response Functions of liquidity measures across the quintile groups.** Displays the impulse response functions of liquidity measures across the Quintile groups over an observation period of 10 days.
(DOCX)

## Acknowledgments

The authors would like to acknowledge helpful suggestions on an earlier version of this paper from the academic editor of the journal, and two anonymous reviewers of the journal who have helped improve the paper significantly.

## Author Contributions

**Conceptualization:** Priyanka Naik, Y. V. Reddy.

**Data curation:** Priyanka Naik.

**Formal analysis:** Priyanka Naik.

**Methodology:** Priyanka Naik, B. G. Poornima.

**Supervision:** Y. V. Reddy.

**Validation:** Priyanka Naik, B. G. Poornima.

**Visualization:** Priyanka Naik, Y. V. Reddy.

**Writing – original draft:** Priyanka Naik.

**Writing – review & editing:** Priyanka Naik, B. G. Poornima, Y. V. Reddy.

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
