## [Decision Letter · Decision Letter 0]

20 Jul 2020

PONE-D-20-17254

Measuring liquidity in Indian stock market: A dimensional perspective

PLOS ONE

Dear Dr. Reddy,

Thank you for submitting your manuscript to PLOS ONE. After careful consideration, I feel that it has merit but does not fully meet PLOS ONE’s publication criteria as it currently stands. Therefore, I invite you to submit a revised version of the manuscript that addresses the points raised during the review process.

It seems that both reviewers agree that the papers could be suitable of publication in the journal after a major revision. The major concern is the contribution of your study in comparison with other previous ones. Please nothe that both reviewers consider that Introduction and conclusion sections should be rewritten.

We look forward to receiving your revised manuscript.

Kind regards,

J E. Trinidad Segovia

Academic Editor

PLOS ONE

Journal Requirements:

Reviewers' comments:

Reviewer's Responses to Questions

**Comments to the Author**

1. Is the manuscript technically sound, and do the data support the conclusions?

Reviewer #1: Partly

Reviewer #2: Yes

2. Has the statistical analysis been performed appropriately and rigorously? 

Reviewer #1: Yes

Reviewer #2: Yes

3. Have the authors made all data underlying the findings in their manuscript fully available?

Reviewer #1: No

Reviewer #2: Yes

4. Is the manuscript presented in an intelligible fashion and written in standard English?

Reviewer #1: No

Reviewer #2: No

5. Review Comments to the Author

Reviewer #1: Review of the paper “Measuring liquidity in Indian stock market: A dimensional perspective

by Y.V. Reddy et al.

1. The manuscript is concerned with measuring liquidity in Indian stock market. It is relevant and within the scope of the journal.

2. However, the manuscript, in its present form, contains several weaknesses. Adequate revisions to the following points should be undertaken in order to justify recommendation for publication.

3. For readers to quickly catch the contribution in this work, it would be better to highlight major difficulties and challenges, and your original achievements to overcome them, in a clearer way in abstract and introduction.

4. The authors must explicitly state the novel contribution of this work, the similarities and the differences of this work with their previous publications.

6. The discussion section in the present form should be strengthened with more details and justifications.

Reviewer #2: I consider the topic to be relevant since it provides interesting insights on the liquidity of Indian Stock Market. Additionally, the methodology used seems to be rigorous enough to back the conclusions presented in the paper. The estimation of widely used measures such as the Amihud Iliquidity ratio among others increases the comparability of the study. Both the abstract and the introduction sections of the paper provide sufficient information about the main research objective and the methodological approach. However, I would suggest to modify the introduction by adding a brief explanation of the concepts introduced in the abstract such as: depth, breadth, tightness, and so on. Moreover, adding more detailed explanation of the main results would be greatly appreciated by the reader since it provides a full picture of the main results, as well as the contribution of the study. Overall, I consider the paper to be a valuable contribution with a sound methodology and well developed theoretical background. My main suggestion would be to review the abstract and introduction sections in terms of quality of written English and ensure they provide all the required information.

6. PLOS authors have the option to publish the peer review history of their article (what does this mean?). If published, this will include your full peer review and any attached files.

Reviewer #1: No

Reviewer #2: No

---

## [Author Response · Author response to Decision Letter 0]

27 Jul 2020

Reviewer #1

1. For readers to quickly catch the contribution in this work, it would be better to highlight major difficulties and challenges, and your original achievements to overcome them, in a clearer way in abstract and introduction.

This study eliminates the ambiguities related to market liquidity by precisely measuring it by using popular and proven liquidity measures. (Included in the manuscript on page no. 2 in Abstract)

Notably, a major challenge while undertaking this study was the selection of appropriate liquidity measures across the four dimensions since the previous literature propounds numerous liquidity measures and their variants. This was resolved by selecting widely acknowledged liquidity measures through extensive analysis of a pool of literary works published on market liquidity. (Included in the manuscript on page no. 4 in Introduction in paragraph 1)

2. The authors must explicitly state the novel contribution of this work, the similarities and the differences of this work with their previous publications.

Moreover, this study offers a novel approach of inclusively studying market liquidity by virtue of widely accepted liquidity measures in an order-driven market system. Further, estimating interdependencies between liquidity dimensions by using daily frequency data and emphasizing the patterns formed thereon is a distinct attempt made with reference to an equity market and will enable traders to effectively optimize their trading strategies. (Included in the manuscript on page no. 5 in Introduction in paragraph 1) 

Similarities and the differences of this work with previous publications are included in the manuscript in Conclusion on page no. 24 in paragraph 2 and on page no. 25 in paragraph 1.

3. The discussion section in the present form should be strengthened with more details and justifications.

Included in the manuscript from page no. 13 to 24 in Results and Discussion

Reviewer # 2

1. I would suggest to modify the introduction by adding a brief explanation of the concepts introduced in the abstract such as: depth, breadth, tightness, and so on.

Additionally, Kyle [5] mentions that liquidity of a market can be understood in terms of three facets namely; the quantity of securities that is traded (depth), the ability of the security prices to quickly recover after a liquidity shock (resiliency) and the costs incurred in trading security (tightness). Furthermore, the time taken to execute a trade (immediacy) and the intensity of trading volume impact on security prices (breadth) is also regarded as the additional facets of market liquidity [6]. (Included in the manuscript on page no. 3 in Introduction in paragraph 2)

2. Moreover, adding more detailed explanation of the main results would be greatly appreciated by the reader since it provides a full picture of the main results, as well as the contribution of the study.

Included in the manuscript in Introduction on page no. 4 in paragraph 2 and on page no. 3 in paragraph 1

3. My main suggestion would be to review the abstract and introduction sections in terms of quality of written English and ensure they provide all the required information.

Included in the manuscript (Refer Abstract and Introduction)

---

## [Decision Letter · Decision Letter 1]

24 Aug 2020

Measuring liquidity in Indian stock market: A dimensional perspective

PONE-D-20-17254R1

Dear Dr. Reddy,

We’re pleased to inform you that your manuscript has been judged scientifically suitable for publication and will be formally accepted for publication once it meets all outstanding technical requirements.

Kind regards,

J E. Trinidad Segovia

Academic Editor

PLOS ONE

Additional Editor Comments (optional):

Reviewers' comments:

Reviewer's Responses to Questions

**Comments to the Author**

1. If the authors have adequately addressed your comments raised in a previous round of review and you feel that this manuscript is now acceptable for publication, you may indicate that here to bypass the “Comments to the Author” section, enter your conflict of interest statement in the “Confidential to Editor” section, and submit your "Accept" recommendation.

Reviewer #1: All comments have been addressed

Reviewer #2: All comments have been addressed

2. Is the manuscript technically sound, and do the data support the conclusions?

Reviewer #1: Partly

Reviewer #2: Yes

3. Has the statistical analysis been performed appropriately and rigorously? 

Reviewer #1: Yes

Reviewer #2: Yes

4. Have the authors made all data underlying the findings in their manuscript fully available?

Reviewer #1: No

Reviewer #2: Yes

5. Is the manuscript presented in an intelligible fashion and written in standard English?

Reviewer #1: Yes

Reviewer #2: Yes

6. Review Comments to the Author

Reviewer #1: The authors have done all commnets on your paper and I believe they can expand this idea in the other financial problems.

Reviewer #2: (No Response)

7. PLOS authors have the option to publish the peer review history of their article (what does this mean?). If published, this will include your full peer review and any attached files.

Reviewer #1: No

Reviewer #2: No

---

## [Editor Report · Acceptance letter]

27 Aug 2020

PONE-D-20-17254R1 

Measuring liquidity in Indian stock market: A dimensional perspective 

Dear Dr. Reddy:

I'm pleased to inform you that your manuscript has been deemed suitable for publication in PLOS ONE. Congratulations! Your manuscript is now with our production department. 

Kind regards, 

on behalf of

Dr. J E. Trinidad Segovia 

Academic Editor

PLOS ONE